# Current Status and Challenges of Safety Education for Children, Including Those Requiring Medical Care, in Japanese General Schools: Focusing on Disasters

**DOI:** 10.3390/children7060065

**Published:** 2020-06-21

**Authors:** Tomoko Yamamoto

**Affiliations:** Department of Music Culture Education, Faculty of Music, Kunitachi College of Music, Tachikawa 1908520, Japan; yamamoto.tomoko@kunitachi.ac.jp; Tel.: +81-42-535-9548

**Keywords:** children requiring medical care, inclusive education, general schools, safety education, Japan

## Abstract

This paper discusses school safety mainly in relation to safety education and examines measures for the comprehensive development of systems to ensure the safety of children at school, including those requiring medical care. The number of children requiring medical care is increasing in general schools following the promotion of inclusive education. The government of Japan has established the School Health and Safety Act and Guidelines on School Curricula. Municipalities have developed education systems that cover the safety education specified in disaster preparedness plans for schools. Safety education has been promoted through problem-oriented and experience-based methods as well as other methods of learning to date. Considering the outcomes of these systems and approaches, it is expected that safety management systems in schools, especially general schools that include children requiring medical care, will also develop in combination with safety education.

## 1. Introduction 

In recent years, there have been many natural disasters in Japan. In 2019, a linear rain band caused localized heavy rainfalls mainly in the Kyushu area, and a typhoon, which was the largest ever in terms of intensity at the time of landing, led to record-breaking rainfall affecting the Kanto area. 

Under these circumstances, schools are adopting disaster safety measures. Further development of safety systems is required in general schools, where an increasing number of children requiring medical care attend under the inclusive education program. 

As part of school safety promotion, disaster safety education has been provided in combination with safety management. In 2012, the year after the Great East Japan Earthquake, the Central Council for Education discussed safety management and safety education, which were summarized in a report entitled: “The Creation of a Plan to Promote School Safety” [1]. In the same year, the Plan to Promote School Safety was established by cabinet decision. Over five years, education for both the acquisition of safety knowledge and the development of safety behaviors was provided. 

With an increase in children requiring medical care attending general or other schools, the Ministry of Education, Culture, Sports, Science, and Technology published a final report of review meetings on medical care provision, including disaster management, in schools in February 2019 [2]. The report discusses “systems to provide medical care in schools” and urges “the establishment of organizational systems in schools”, emphasizing that it is necessary for each school to adopt safety assurance measures, including emergency management measures, individualized manuals for crisis management, and systems to contact related (welfare/medical) institutions based on guidelines or similar standards established by the relevant departments of education, in addition to documenting these items. It highlights the importance of establishing means of communication for emergencies. Furthermore, as a “disaster management measure”, it indicates the necessity of consulting with parents about the preparation and storage of medical materials, medical devices, and emergency food in schools to provide appropriate medical care for children requiring medical care according to their conditions even during disasters. Specifically, in schools admitting children requiring medical care using medical devices, such as mechanical ventilation systems, the availability of power sources, duration of batteries for medical devices, and measures to be adopted when the electricity fails should be confirmed in advance by those working for schools (including school doctors, medical care advisers, and nurses contracted by schools) and parents. Concerning the management of disasters occurring during the school commute, such as when traveling from home to school by bus, the report also recommends sufficiently confirming measures to be adopted in emergencies and systems to collaborate and cooperate with medical institutions. 

Thus, schools have adopted approaches for disaster safety assurance that combine safety management and safety education. However, measures to ensure the safety of children requiring medical care during disasters are still decided within a framework of safety management by each school, those working for schools (including school doctors, medical care advisers, and nurses contracted by schools), medical institutions, related medical/welfare institutions, and parents. 

Previous studies on safety management systems in Japanese schools, especially in general schools, confirmed the roles to be played during disasters [3], discussed various appropriate methods for collaboration in such cases [4], and examined education to promote disaster preparedness and ability development for crisis avoidance [5]. On the other hand, ensuring the safety of children in general schools, including those requiring medical care, is also a new topic for discussion in Japan, where home care for children requiring medical care is being promoted. 

Now, promoting the comprehensive development of systems to implement disaster management measures in schools with these children while considering current education systems and curricula for school safety is an important challenge. This paper discusses school safety mainly in relation to safety education and examines challenges related to safety management systems in schools, especially in general schools, for children, including those requiring medical care. Although medical and education systems vary among countries, Japan is not the only country with many natural disasters; therefore, clarifying the current status and challenges of safety education in Japan may have international significance. 

## 2. Materials and Methods 

The purpose of this paper is to clarify the current status of safety education in general schools, focusing on disasters, as a basis for the development of safety assurance measures for children at school, including those requiring medical care. 

In Japan, systems for disaster education at school have been established, and education related to disaster safety is provided based on them. The following section first outlines the systems for disaster education at school and then analyzes the pioneering educational approaches adopted mainly in elementary schools to discuss the outcomes and challenges of these educational approaches based on the current education system. 

## 3. Results 

### 3.1. National Systems for Disaster Preparedness Education in Schools

#### 3.1.1. School Health and Safety Act

The School Health and Safety Act (final amendment in 2015; act number 46) specifies systems for disaster preparedness education in schools. 

Articles 26 to 30, Chapter 3, of this act define school safety. 

For example, the school safety-related responsibilities of school founders, such as the government of a municipality, should organize and improve facilities, equipment, and management and operation systems in the school or adopt other necessary measures not only to appropriately manage crises among students caused by disasters but also to prevent them. When any facility- or equipment-related difficulties in ensuring the safety of students have been identified in a school, the principal of the school should adopt measures for improvement with no delay or, when it is infeasible to adopt those measures, inform the school founder of this (Article 28). 

To ensure the safety of students at school, Crisis Management Guidelines that specify measures and procedures to be adopted by the staff of each school in the event of a crisis should be created (Article 29). These measures include providing support for students and others for whom disasters have caused psychological trauma or other mental/physical disorders. The principal of each school is also urged to improve the recognition of the Crisis Management Guidelines among school staff and adopt necessary measures for the staff to appropriately manage the situation in the event of a crisis. 

To ensure the safety of students at school, school safety plans should be created and implemented in connection with disaster preparedness education (Article 27). These plans also cover safety inspections, school staff training, other school safety-related issues, guidance on safety in school, and other daily activities for children. 

Thus, as shown in Figure 1, the School Health and Safety Act specifies the importance of combining safety management and safety education as the foundation of safety assurance in schools.

The act emphasizes the necessity of promoting collaboration with parents and, according to the actual situation of the school, among related institutions, groups, residents, and others involved in activities to ensure the safety of their communities (Article 30). 

#### 3.1.2. Guidelines on School Curricula

The Guidelines on School Curricula also contain provisions on disaster preparedness education. 

The government created these guidelines as standards for education courses to be organized by each school with the aim of ensuring a certain level of education throughout Japan based on the School Education Act (enacted in 1947; act number 26). Its revised version, which is scheduled to be adopted in elementary schools from FY2020, specifies safety education-related provisions in “General rules and related courses” [6]. These measures concerning disaster preparedness and accident prevention in communities aim at improving the quality of teaching in order to improve children’s abilities to safely act and aid in the safety of others. For example, these measures promote children learning about the geographic structure of the islands of Japan and the mechanisms of earthquakes. Furthermore, disaster preparedness education is being provided through collaboration/cooperation with city offices and fire stations as part of comprehensive learning. 

### 3.2. Status of Disaster Preparedness Education in General Schools and Communities

#### 3.2.1. Creation of Disaster Preparedness Plans for Schools

The departments of education create disaster preparedness plans for schools based on municipal disaster preparedness plans that specify measures to be adopted in each school during disasters [7]. 

As shown in Figure 2, there are three “purposes” of education defined in the disaster preparedness plans for schools: the “development of disaster preparedness literacy>”, “development of abilities for self- and mutual help”, and “development of insights into living as a human and the dignity of life”. 

The “development of disaster preparedness literacy” includes understanding the mechanisms of natural disasters, past disasters, and disaster preparedness systems; accurately recognizing possible critical situations during disasters; adopting daily preparedness measures; and acquiring abilities to quickly adopt actions to ensure one’s own safety based on appropriate judgment. 

The “development of abilities for self- and mutual help” includes acquiring abilities to make appropriate judgments and adopt appropriate actions independently; developing attitudes and abilities to willingly participate in volunteer activities, while helping others and addressing difficult situations during disasters; and understanding basic items regarding disasters/disaster preparedness. 

Finally, the “development of insights into living as a human and the dignity of life” includes developing insight into the dignity of life and consideration of others, the importance of protecting one’s life, and empathy towards disaster victims.

The first purpose, “Development of disaster preparedness literacy”, focuses on abilities to make appropriate judgments about safety, accurately recognize crises, and understand disaster preparedness systems. 

To fulfill this purpose, “education content”, represented by “promoting disaster preparedness education according to developmental stages” and “integrating learning and examining/adopting various curricula” in addition to the following practical educational approaches has been adopted. These approaches include “project learning”: clarifying wishes, setting themes, developing learning strategies, collecting information, and achieving goals; “crisis prediction training”: preparing sheets to simulate disasters, forming teams to discuss methods to detect and avoid crises, setting behavioral goals, and adopting these behaviors; and “problem-solving discussions on disasters using maps”: discussing and sharing challenges and necessary measures in communities using school zone maps. 

#### 3.2.2. Status of Safety Education in General Schools

##### (1) Problem-Oriented Learning in Daily Life

There are two types of safety education provided in general schools: (1) safety education focusing on problems faced by disaster victims in daily life, which is being provided in the Great East Japan Earthquake-affected areas [8]; and (2) safety education to address problems that may occur in daily life [9]. 

In the first type, learning materials developed with children affected by the disaster are used [10]. For example, a textbook for first- and second-grade students at elementary school has a cover with an illustration of “the sea”. The first page is entitled: “3.11—We will never forget the Great East Japan Earthquake”, and the following pages address different themes, including “learning about disasters”, “receiving mental care”, “thinking about the way of living”, “helping each other, living together”, “learning about public support and preparedness”, and “protecting oneself”. In the “protecting oneself” section, example actions to be taken by children to address situations, such as “near the sea” and “when a black cloud comes”, in addition to when they are “at school”, “at home”, and “outside”, are presented with effective illustrations. 

The findings on school safety obtained through the Great East Japan Earthquake are being shared throughout Japan by “handing down lessons” through collaboration/cooperation with elementary schools in these areas. 

In the second type, safety education to resolve specific problems, such as “difficulty returning to home/handing over”, “evacuating under a tsunami”, “shelter management”, and “disaster preparedness volunteer activities”, is being provided as part of disaster preparedness education for school safety assurance, focusing on the dignity of life. For example, community-based elementary schools promote learning about “shelter management”-related problems. 

The development of a way of thinking to adopt appropriate actions during disasters is another focus of this type of safety education [11]. For example, all elementary school pupils, including first-graders, discuss together how to clarify, judge, and address specific situations, such as a fire occurring in an adjacent house during a disaster. Pupils are supported to develop appropriate thoughts to protect their own and others’ lives during each type of disaster and acquire necessary knowledge and skills to adopt these measures individually and through collaboration with others. 

As a recent trend, some elementary schools conduct evacuation drills without notice as an educational approach to help pupils utilize the outcomes of their daily learning. In such a drill, first-grade elementary school pupils in a classroom tried to protect each other by ordering each other to get under the desk, although they experienced anxiety in the absence of their teacher [12]. 

##### (2) Experience-Based Learning through Collaboration/Cooperation with Communities 

In general schools, experience-based learning through collaboration/cooperation with communities is also being promoted. A typical example is disaster preparedness drills with communities. 

The author participated in one of the annual disaster preparedness drills conducted by an elementary school with its community. During the drill, more than 400 community residents gathered in a place designated by the residents’ association and moved to the schoolyard as a shelter from early in the morning. Teachers and students of the school also participated in the drill. District-based groups of participants gathered in the schoolyard and then learned about disaster preparedness measures adopted in the community. They also experienced putting out a fire and an earthquake simulation. The drill aimed to help participants acquire knowledge and skills for disaster preparedness. It also provided an opportunity to learn methods to form a municipal or school organization and operate it through collaboration/cooperation. 

Some elementary schools also utilize disaster preparedness drills for shelters by experience-based learning [13]. In such learning, while supervised by school teachers, students organize shelters through collaboration/cooperation with community residents and subsequently stay overnight in gymnastics halls simulating shelters. To experience life in shelters, they also perform the following activities: developing strategies to survive with only two 500 mL bottles of water/person until the next morning, consuming pregelatinized rice (with other ingredients) for dinner and breakfast as an emergency food, living without a power supply, sleeping without a bed, experiencing an aftershock early in the morning, and solving post-disaster problems without clear solutions. The aim of such learning is to help children prepare themselves for natural disasters that may suddenly occur in Japan, focusing on the ability to protect themselves and others in communities. 

#### 3.2.3. Requirements to Ensure the Safety of Children, Including Those Requiring Medical Care, at School 

According to Ken Kasai (TEAM Bousai—Disaster Preparedness—Japan), providing sufficient medical care/materials despite difficulties such as destroyed facilities, roads, and disaster-affected personnel; maintaining appropriate hygienic environments, including bathing support; and preventing infection are challenges to ensuring the safety of children requiring medical care [14]. 

In some cases, special measures may also be required immediately after the occurrence of a major disaster, such as moving mechanical ventilation systems or other medical devices and ensuring sufficient sources of power for these devices. Therefore, safety education for children in general schools should also address new challenges specific to children requiring medical care, represented by the mobility of medical devices and availability of sufficient power supplies, and develop comprehensive systems and approaches to ensure the safety of these children in general schools, as shown in Figure 3. Furthermore, as difficulties vary among children requiring medical care, optimal solutions should be examined with a view to covering issues specific to these children in safety education. This is also important when evaluating the outcomes. 

Finally, systems for nurses who provide medical care to schools to adopt necessary measures, such as safety management and specialized support, are being established based on the Final Report of Review Meetings on Medical Care in Schools. 

## 4. Discussion 

According to the School Health and Safety Act and Guidelines on School Curricula, national systems for safety assurance in schools should address safety education in addition to safety management. Concerning safety education, disaster preparedness education provided in communities aims to develop an understanding of disasters and disaster preparedness and foster “disaster preparedness literacy”, including the judgment needed to ensure one’s own and others’ safety. 

In general, safety education in schools adopts numerous approaches to resolving challenges. An example of this is experience-based learning, where children acquire knowledge and skills by learning about problems experienced in past cases. Each of them determines solutions to problems that may occur but will be difficult to resolve in the future. The children then share this process and results with others and make a final decision in a group. Through such learning, children learn about different issues from those they have experienced to those that they may encounter in the future. 

Furthermore, to establish inclusive safety education, general schools should examine measures to resolve challenges specific to children requiring medical care, such as the mobility of medical devices and the availability of sufficient power supply to use them upon deliberation with the children. In particular, when elementary schools are used as temporary designated shelters or there is difficulty living at home, safety education is also expected to address issues related to daily life, including the maintenance of eating, toileting, hygiene care, and other related activities; medical care; and materials/human resources. 

Thus, school safety measures should be implemented combining safety management and safety education. The Final Report of Review Meetings on Medical Care in Schools specifies the details of safety management to be performed by each school, those working for schools (including school doctors, medical care advisers, and nurses contracted by schools), medical institutions, related (medical/welfare) institutions, and parents. In order to develop safety assurance measures for children including those requiring medical care in general schools, it may be necessary to promote comprehensive safety assurance systems and education covering both safety education and safety management. 

## 5. Conclusions 

This paper discusses challenges related to safety education and safety management in relation to measures to ensure the safety of children, including those requiring medical care, during disasters. 

Comprehensive approaches combining safety management and safety education are required to promote children’s disaster preparedness while protecting them during disasters. Safety education provided in communities also aims to foster their abilities to think and act appropriately and independently in the process of developing solutions to individual disaster-related problems. 

To ensure the safety of children, including those requiring medical care, systems and safety education combined with safety management should be examined from the perspective of collaboration/cooperation between schools and those working for schools (including school doctors, medical care advisers, and nurses contracted by schools), medical institutions, related medical/welfare institutions, and parents. 

## Figures and Tables

**Figure 1 children-07-00065-f001:**
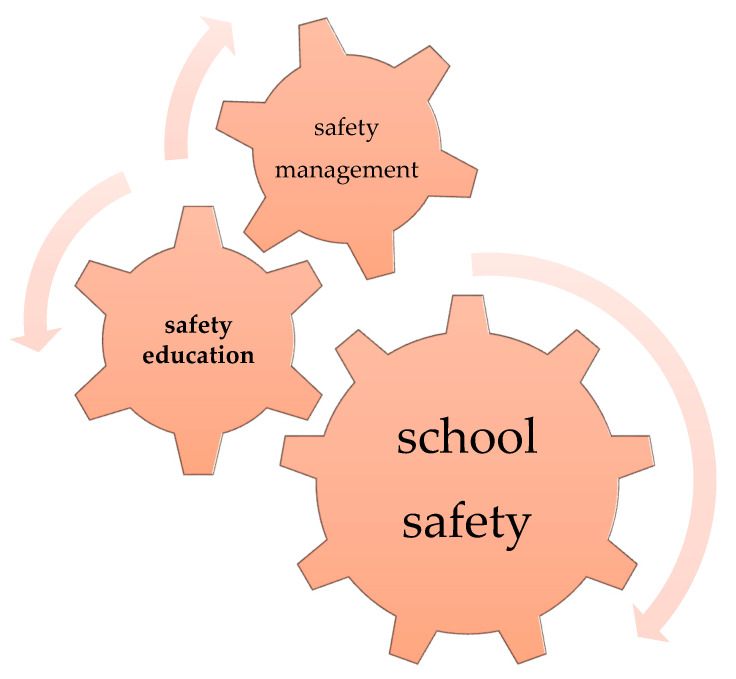
Basics of safety assurance in schools based on the school health and safety act.

**Figure 2 children-07-00065-f002:**
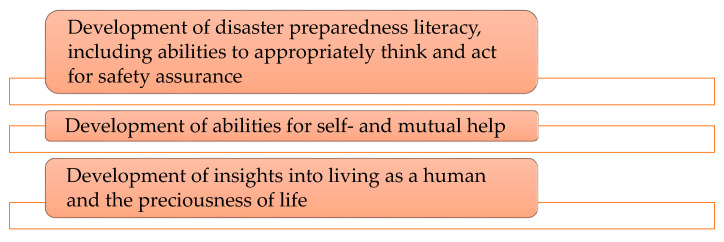
Purposes of education specified in disaster preparedness plans for schools.

**Figure 3 children-07-00065-f003:**
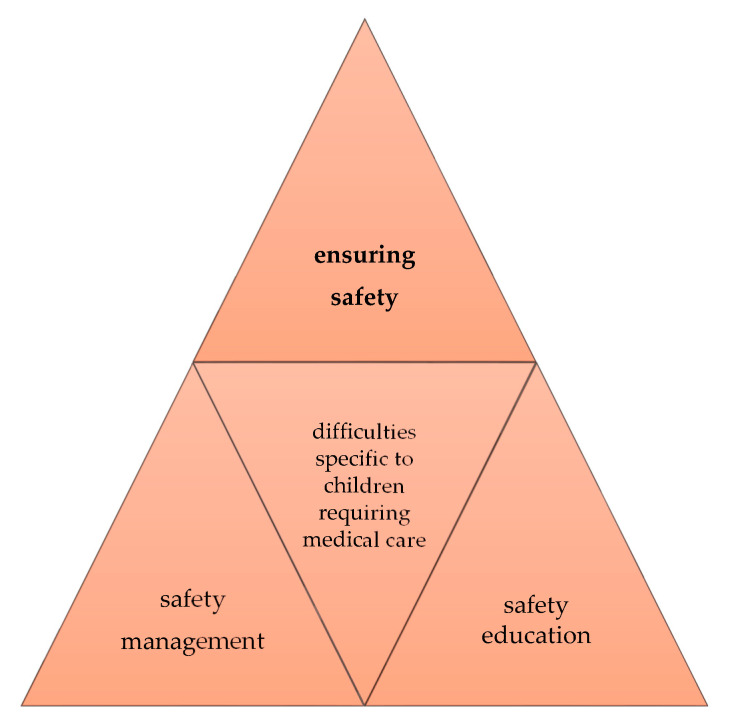
Safety assurance measures for children, including those requiring medical care, in general schools.

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
