# Peer review of "Current Status and Challenges of Safety Education for Children, Including Those Requiring Medical Care, in Japanese General Schools: Focusing on Disasters"

_children, 2020, doi:10.3390/children7060065_

Round 1

Reviewer 1 Report

The manuscript presented raises a topic that may be of interest; However, it has been approached from a perspective that does not represent a great contribution to the scientific community.
Firstly, the manuscript is presented as an empirical research (methodology, results, discussion), however it is a systematic review, which does not follow any methodology in the process of searching, selecting and analyzing the sources.
In this sense, the methodology is out of place and does not include relevant information. Likewise, in the results section there are no results in themselves, but information extracted from other sources. The discussion section also does not respond to what is meant by discussion of the results among the scientific community: there is no contrast between the results of the research and the other studies or reports. This section presents only conclusions drawn from the bibliographic review process.
Lastly, it is a bibliographic review manuscript but there is no in-depth review of the subject at the national and international level. It only includes 14 sources, which is more than scarce in a scientific work.

Author Response

Dear Reviewer 1, 

     Thank you for the thoughtful and constructive feedback you provided regarding my manuscript. I have listed my responses to each comment. 

(about the format of the manuscript) 

    Following your advice, I revised the description of the methods. I have reformulated mu manuscript so as to report the outcomes and challenges of these pioneering educational approaches being implemented based on relevant systems more clearly. 

(about the regulations for references) 

     I  corrected the references thanks to your advise. 

(about the paper's materials in English) 

     Safety education for children requiring medical care in Japanese general schools is a new topic being introduced for research, and there are no papers addressing this in Pub Med. However, as you advised, I have added an English references 4) discussing diverse approaches to collaboration. 

Reviewer 2 Report

Review of the commentary entitled "Current Status and Challenges of Safety Education for Children, Including Those Requiring Medical Care, in Japanese General Schools: Focusing on Disasters".

The manuscript itself serves as the author's opinion on educational security in Japanese general schools.
My suggestions are related to the format of the manuscript submitted. It is not an investigation so it is not very well understood to introduce the sections of material and methods or results. I would therefore strongly recommend that the structure of the manuscript be reformulated.
Furthermore, the references are not adapted to the regulations.

Author Response

Dear Reviewer 2, 

     Thank you for the thoughtful and constructive feedback you provided regarding my manuscript. I have responded to the comments as follows. 

(about the format of the manuscript) 

     Following your advice, I revised the description of the methods. I have reformulated my manuscript so as to report the outcomes and challenges of these pioneering educational approaches being implemented based on relevant systems more clearly. 

Reviewer 3 Report

In the process of the review of this draft, the content of education for Japanese schools' disaster management was interesting.

[1] Overall, the same phrases are being repeated several times, which are likely to be deleted: for example, the children requiring medical care, or children including those requiring medical care; those working for schools (including school doctors, medical care advisers, and nurses contracted by schools);

[2] Through this paper, the author wanted to share the knowledge how to educate students about natural disasters. Unfortunately, however, all 14 references are in Japanese. If possible, the authors would like to provide literature or websites that allow them to search the paper's materials in English.

[3] Trivial one: on page 3, line 91, there is a word 'founder'. What does the founder mean? If the founder died because of a long history of school, who is responsible for the school facilities? 

Author Response

Dear Reviewer 3, 

     Thank you for the thoughtful and constructive feedback you provided regarding my manuscript. I have listed my responses to each comment. 

(about the same phrases) 

     I deleted the repeated phrases. 

(about the paper's materials in English) 

     Safety education for children requiring medical care in Japanese general schools is a new topic being introduced for research, and there are no papers addressing this in Pub Med. However, as you advised, I have added an English reference 4) discussing diverse approaches to collaboration. 

(about the word 'founder') 

     Thank you for your suggestion. To help understand the meaning of "school founder", I have added the phrase "such as the government of a municipality" to this sentence.  

Round 2

Reviewer 1 Report

I maintain the opinion reflected in the first review. The article does not show empirical research and therefore should not follow that structure. It is not very rigorous and includes few sources in the case of a review of the state of affairs. The authors have not carried out the proposed restructuring, so from my point of view it does not meet the requirements for publication.

Author Response

Dear Reviewer 1, 

Thank you for the thoughtful and constructive feedback you provided regarding my manuscript. I have listed my responses to each comment. 

Point 1: Firstly, the manuscript is presented as an empirical research (methodology, results, discussion), however it is a systematic review, which does not follow any methodology in the process of searching, selecting and analyzing the sources. In this sense, the methodology is out of place and does not include relevant information. Likewise, in the results section there are no results in themselves, but information extracted from other sources. The discussion section also does not respond to what is meant by discussion of the results among the scientific community: there is no contrast between the results of the research and other studies or reports. This section presents only conclusions drawn from the bibliographic review process. 

Response 1: Following your advice, I revised the description of the methods. I have reformulated my manuscript so as to report the outcomes and challenges of these pioneering educational approaches being implemented based on relevant systems more clearly. 

Point 2: Lastly, it is a bibliographic review manuscript but there is no-depth review of the subject at the national and international level. It only includes 14 sources, which more than scare in a scientific work. 

Response 2: Safety education for children requiring medical care in Japanese general schools is a new topic being introduced for research, and there are no papers addressing this in Pub Med. However, as you advised, I have added an English reference 4) discussing diverse approaches to collaboration.  
